# Schizotrophic *Sclerotinia sclerotiorum*-Mediated Root and Rhizosphere Microbiome Alterations Activate Growth and Disease Resistance in Wheat

Binnian Tian,[a,b,d] Zheng Qu,[b,c] Mirza Abid Mehmood,[b,c] Jiatao Xie,[b,c] Jiasen Cheng,[b,c] Yanping Fu,[c] Daohong Jiang[b,c]

[a]College of Plant Protection, Southwest University, Chongqing, China
[b]State Key Laboratory of Agricultural Microbiology, Huazhong Agricultural University, Wuhan, China
[c]The Provincial Key Lab of Plant Pathology of Hubei Province, Huazhong Agricultural University, Wuhan, China
[d]Key Laboratory of Agricultural Biosafety and Green Production of Upper Yangtze River, Ministry of Education, Southwest University, Chongqing, China

**ABSTRACT** *Sclerotinia sclerotiorum*, a widespread pathogen of dicotyledons, can grow endophytically in wheat, providing protection against *Fusarium* head blight and stripe rust and enhancing wheat yield. In this study, we found that wheat seed treatment with strain DT-8, infected with *S. sclerotiorum* hypovirulence-associated DNA virus 1 (SsHADV-1) and used as a "plant vaccine" for brassica protection, could significantly increase the diversity of the fungal and bacterial community in rhizosphere soil, while the diversity of the fungal community was obviously decreased in the wheat root. Interestingly, the relative abundance of potential plant growth-promoting rhizobacteria (PGPR) and biocontrol agents increased significantly in the DT-8-treated wheat rhizosphere soil. These data might be responsible for wheat growth promotion and disease resistance. These results may provide novel insights for understanding the interaction between the schizotrophic microorganism and the microbiota of plant roots and rhizosphere, screening and utilizing beneficial microorganisms, and further reducing chemical pesticide utilization and increasing crop productivity.

**IMPORTANCE** Fungal pathogens are seriously threatening food security and natural ecosystems; efficient and environmentally friendly control methods are essential to increase world crop production. *S. sclerotiorum*, a widespread pathogen of dicotyledons, can grow endophytically in wheat, providing protection against *Fusarium* head blight and stripe rust and enhancing wheat yield. In this study, we discovered that *S. sclerotiorum* treatment increased the diversity of the soil fungal and bacterial community in rhizosphere soil, while the diversity of the fungal community was obviously decreased in the wheat root. More importantly, the relative abundance of potential PGPR and bio-control agents increased significantly in the *S. sclerotiorum*-treated wheat rhizosphere soil. The importance of this work is that schizotrophic *S. sclerotiorum* promotes wheat growth and enhances resistance against fungal diseases via changes in the structure of the root and rhizosphere microbiome.

**KEYWORDS** Schizotrophic microorganism, *S. sclerotiorum*, wheat, microbiome, PGPR, biocontrol agents

The plant microbiome is often referred to as the host's second or extended genome, which comprises various microorganisms, including bacteria, fungi, viruses, and oomycetes (1). Plant roots are the primary sites of nutrient import and organic molecule export, which provide carbon and energy sources to nearby microorganisms, and the rhizosphere has a higher microbe number than bulk soils (2, 3). In this process, the root-associated microbiota plays a major role in plant health and productivity by improving nutrient availability, suppressing phytopathogens, and promoting growth by influencing plant hormone

Address correspondence to Daohong Jiang, daohongjiang@mail.hzau.edu.cn.

The authors declare no conflict of interest.

pathways (4). In return, the host plant delivers habitation and a constant supply of energy and carbon sources to the microbiota (5). Therefore, the coordination between the root microbiota and the host plant is essential for plant growth in natural environments.

Nitrogen is considered a critical limiting nutrient element for plant growth and is mainly present as nitrate, ammonium, and organic nitrogen in the soil (6). Despite plants using a wide range of N forms, research on plant N nutrition has had a strong focus on inorganic N forms (7). Different forms of nitrogen can be metabolized by bacteria, and this might affect the efficiency of plant nitrogen absorption, as plants have a preference for inorganic nitrogen (ammonium [$NH^{4+}$] and nitrate [$NO^{3-}$]) rather than organic nitrogen (8, 9). Among these, nitrifying microorganisms play an essential role by performing the stepwise aerobic oxidation of ammonia to nitrate. Nitrification is mediated by the ammonia-oxidizing bacteria or archaea, which operate in a tight interplay with nitrite-oxidizing bacteria (10).

*Sclerotinia sclerotiorum* (Lib.) de Bary, a notorious fungal phytopathogen, destroys many economically important dicotyledonous crops, including numerous leguminous and brassicaceous crops (11, 12). Previously, we proved that the systemically endophytic capability of a mycovirus-mediated hypovirulent strain, DT-8, and DT-8VF, a virus-free virulent derivative of DT-8 of *S. sclerotiorum* promotes wheat growth and enhances resistance against *Fusarium* head blight (FHB) and stripe rust. Moreover, we demonstrated that the beneficial effect of *S. sclerotiorum* on wheat was not strain dependent and did not depend on full virulence on susceptible hosts (13). Further, we assumed that the rhizosphere soil and root microbiome of wheat might also play roles in growth promotion and disease resistance.

In this study, the effects of strain DT-8 on the root and rhizosphere soil microbiota in wheat at the initial bloom stage were determined by 16S rRNA and internal transcribed spacer (ITS) sequencing techniques. The correlation between yield enhancement and disease resistance and the microbial community in wheat root and rhizosphere soil were explored.

## RESULTS

***S. sclerotiorum* carries out endophytic growth beneficially in wheat.** To understand the specific effects of strain DT-8 of *S. sclerotiorum* on wheat growth, DT-8-treated and nontreated control wheat plants were planted in a natural field located at Ezhou City in Hubei province, China, in late October 2018. DT-8-treated plants were more robust than nontreated control plants (Fig. 1A), root vigor was significantly greater at the anthesis stage (Fig. 1B), and the nitrogen content in flag leaves was also significantly higher (Fig. 1C). At the same stage, there were no significant differences in the total nitrogen and available nitrogen content in the soil among the DT-8-treated plot and the nontreated control plot (Fig. 1D and E).

To explore the variation of microbiota, wheat root and corresponding rhizosphere soil samples were collected from DT-8-treated and control wheat groups at the initial bloom stage. For the wheat root and rhizosphere soil samples collected from the field, the coat protein gene (*CP*) of *S. sclerotiorum* hypovirulence-associated DNA virus 1 (SsHADV-1) was detected in all root samples taken from the DT-8 treatment group before sequencing, whereas it was not detected in any samples of the nontreatment control plants (see Fig. S1 in the supplemental material). The results indicated that *S. sclerotiorum* strain DT-8 colonized the wheat root in the field after DT-8 treatment.

**Microbiome data acquisition and statistics.** We characterized the multikingdom microbial consortia along the rhizosphere-root by simultaneous DNA amplicon sequencing of the 16S rRNA gene targeting regions V5 to V7, using primers 799F and 1193R, and the fungal, internal transcribed spacer (ITS) regions ITS1 to ITS2, followed by Illumina sequencing.

A total of 1,047,729 high-quality reads from the ITS regions were identified among 20 samples (average, 52,386; range, 43,263 to 63,214 reads per sample), after removing the adaptor sequences and low-quality reads. After removing chimeric and organelle sequences, the high-quality reads were analyzed with USEARCH to produce 1,259 operational taxonomic units (OTUs). Referring to the rarefaction curves (Fig. S2A), the

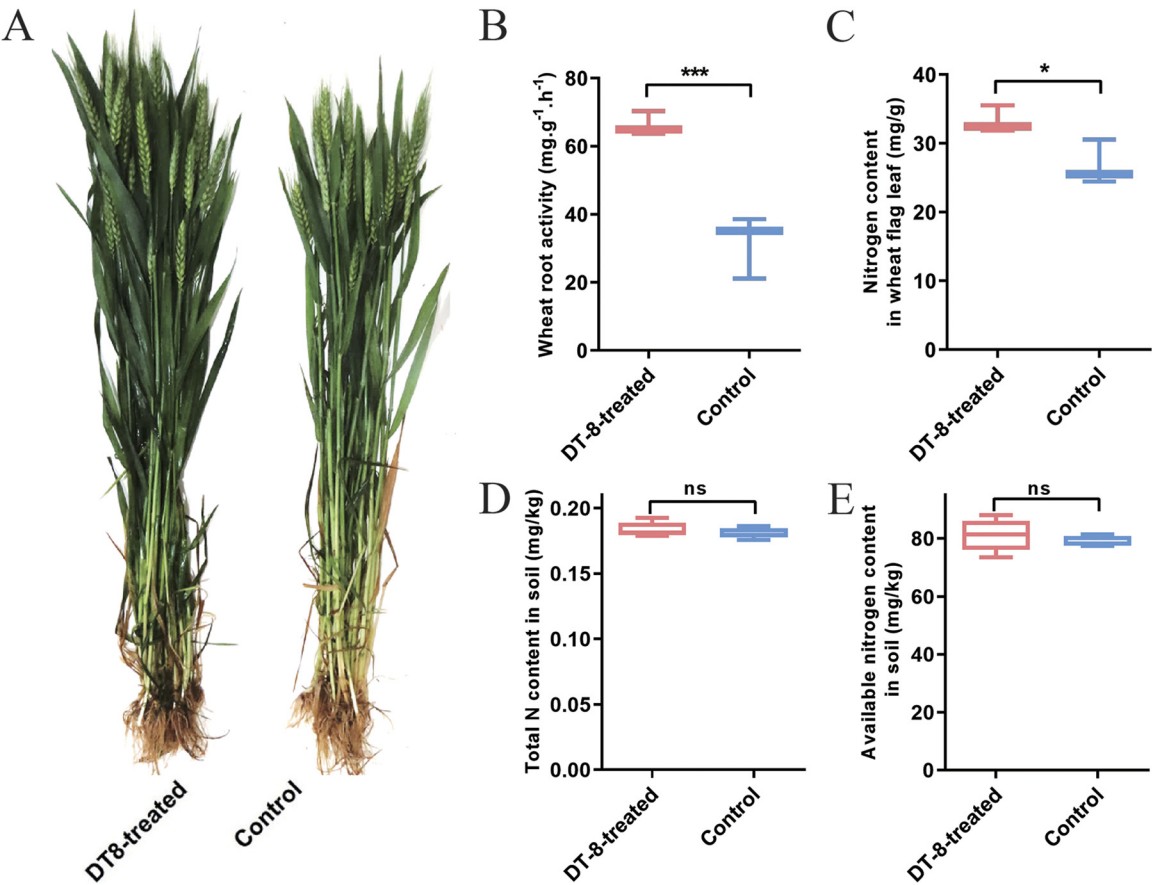

**FIG 1** *S. sclerotiorum* promotes wheat growth. (A) Representative image of wheat plants treated with strain DT-8 in the field at the anthesis stage. (B) Detection of wheat root vigor at the anthesis stage (*t* test, $P < 0.001$). (C) Detection of the nitrogen content in wheat flag leaves at the anthesis stage (*t* test, $P < 0.05$). (D and E) The total nitrogen and available nitrogen content in the soil. Five replicates for the wheat samples and the soil samples were used. *, $P < 0.05$; ***, $P < 0.001$.

data set was normalized to the lowest number of read counts (43,263 reads). For bacterial communities, 558,720 high-quality reads were identified among 20 samples (average, 27,936; range, 23,171 to 35,397 reads per sample), after removing the adaptor sequences and low-quality reads. A total of 3256 OTUs were produced with USEARCH. Referring to the rarefaction curves (Fig. S2B), the data set was normalized to the lowest number of read counts (23,171 reads). The raw data used in this study are deposited in the NCBI GEO database with BioProject no. PRJNA601289 and PRJNA545802.

**Diversity analyses of bacterial and fungal communities. (i) $\alpha$-diversity.** To estimate the differences in the alpha diversity, Shannon's index, the Chao1 index, and the Simpson index were measured, and the results indicated a significant difference between DT-8-treated and control plants. For fungal communities, Shannon's and the Chao1 and Simpson diversity indices exhibited an increase in the rhizosphere soil of DT-8-treated plants compared with the control, whereas there was greatly decreased diversity in the roots (Fig. 2A to C). For bacterial communities, Shannon's and the Chao1 diversity indices exhibited increased diversity in the rhizosphere soil of DT-8-treated plants compared with the control; however, the diversity had no significant variation in the roots (Fig. 2D to F). These results indicate that *S. sclerotiorum* treatment significantly increased the fungal and bacterial diversity index in wheat rhizosphere soil but decreased the fungal diversity in roots.

**(ii) $\beta$-diversity.** Nonmetric multidimensional scaling (NMDS) was applied using unweighted UniFrac distance to identify the community composition of all samples. The composition of the fungal and bacterial microbiota of rhizosphere soil and roots

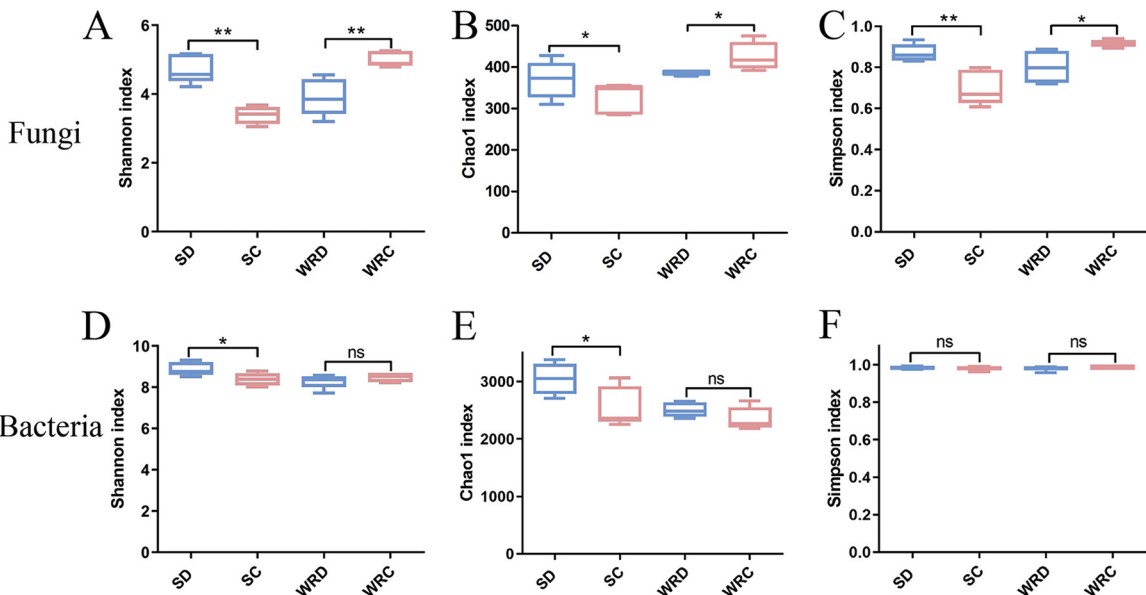

**FIG 2** (A to H) $\alpha$-Diversity of the fungi (A to D) and bacteria (E to H) in wheat communities. $\alpha$-Diversity estimates of the fungi or bacteria in rhizosphere soil and root samples. SD, DT-8-treated wheat plant rhizosphere soil; SC, nontreated wheat plant rhizosphere soil; WRD, DT-8-treated wheat plant root; WRC, nontreated wheat plant root. Five replicates were included for the rhizosphere soil and root samples. *, $P < 0.05$; **, $P < 0.01$.

differed in the DT-8-treated and control groups. The first axis revealed that the fungal and bacterial communities within root samples were different from the corresponding rhizosphere soil samples regardless of sampling in the DT-8-treated or control groups. The second axis revealed that fungal communities of rhizosphere soil and root samples formed two distinct clusters by DT-8 treatment (Fig. 3A). Meanwhile, the bacterial communities of both rhizosphere soil and root samples formed two separated clusters by DT-8 treatment (Fig. 3B). The consistency of the result was also confirmed using the analysis of hierarchical clustering of the samples based on Bray-Curtis dissimilarity (Fig. 3C and D). Both bacterial and fungal samples separating across the second principal coordinate indicated that the source of variation in microbial communities in the root and rhizosphere soil of wheat plants treated with *S. sclerotiorum* was the DT-8 treatment.

**Fungal and bacterial community composition.** To understand the exact composition of microbiota in different samples, a further analysis was done on the individual fungal and bacterial phyla. The fungal and bacterial phyla contributing >90% in relative abundance were classified as "core" phyla, while others were classified as "low abundance." Moreover, the communities that could not be classified within any phyla were grouped and placed in the category "unclassified." The differences in fungal and bacterial communities between DT-8-treated and control samples were significant and detectable at the phylum level. For fungal communities, these phyla, such as Ascomycota, Olpidiomycota, Basidiomycota, Entomophthoromycota, Rozellomycota, Mortierellomycota, and Chytridiomycota, were core phyla in all samples. In rhizosphere soil, Chytridiomycota and Glomeromycota were present at higher relative abundance in DT-8-treated plants than in the control, whereas only one phylum, Entomophthoromycota, was found with a higher relative abundance in the control (Fig. 4A and Table S1). For bacterial communities, such as *Proteobacteria*, *Actinobacteria*, *Bacteroidetes*, *Firmicutes*, *Gemmatimonadetes*, *Acidobacteria*, and *Verrucomicrobia*, were core phyla. In rhizosphere soil, dominant members of the DT-8-treated samples were *Verrucomicrobia*, *Planctomycetes*, *Nitrospirae*, *Gemmatimonadetes*, *Bacteroidetes*, *Latescibacteria*, *Acidobacteria*, *Entotheonellaeota*, and *Thermotogae*. In the root, *Firmicutes*, *Cyanobacteria*, and *Gemmatimonadetes* were present at a higher relative abundance in DT-8-treated plants than in the control (false-discovery rate [FDR] adjusted $P < 0.05$, Wilcoxon) (Fig. 4B and Table S2).

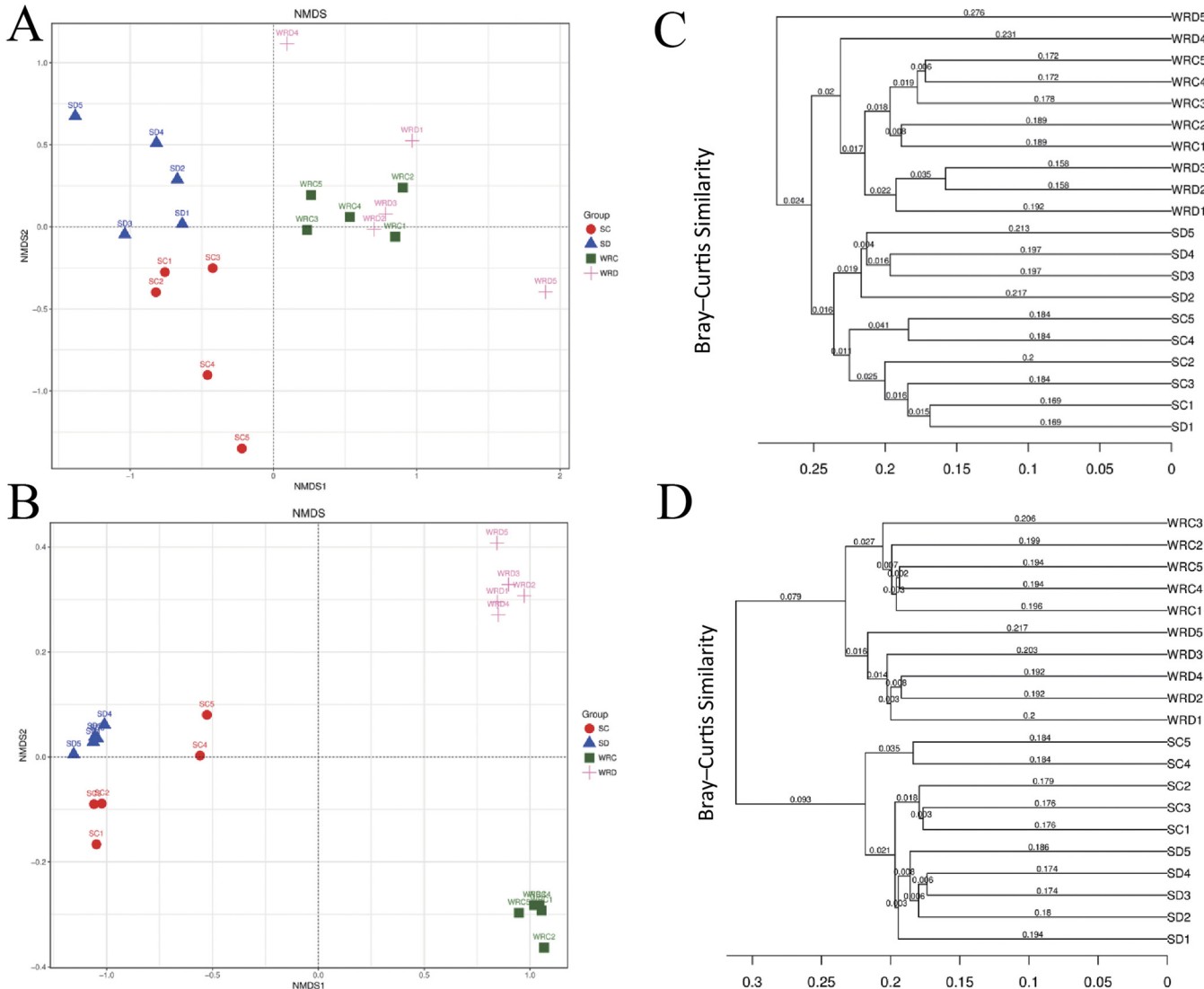

**FIG 3** β-Diversity (NMDS) of the fungi and bacteria in wheat rhizosphere soil and root communities. (A and B) Nonmetric multidimensional scaling (NMDS) of fungi (stress = 0.079) (A) and bacteria (stress = 0.016) (B) communities. (C and D) Hierarchical clustering (group average linkage) of the fungus (C) and bacterium (D) samples based on Bray-Curtis similarity. NMDS and hierarchical clusters were based on 5 biological replicates (rhizosphere soil and root samples).

Pairwise comparisons were conducted to check the microbial abundance using a zero-inflated log-normal model implemented in the MetagenomeSeq package (FDR-corrected $P < 0.05$). *S. sclerotiorum* significantly altered the fungal and bacterial communities at the genus level in roots and the corresponding rhizosphere soil of DT-8-treated wheat compared with controls. In the case of fungi, when comparing the abundance of genera identified in the DT-8-treated wheat with the control, a total of 18 differentially abundant fungal genera were observed in wheat roots, including 2 genera with increased abundance, and 16 genera had decreased abundance in the roots of DT-8-treated wheat. *Rhizoctonia* and *Sclerotinia* showed higher abundance in the roots of the DT-8-treated wheat and lower abundance in the control. In contrast, dominant members of the nontreated samples were *Cyphellophora*, *Exserohilum*, *Dinemasporium*, *Neodevriesia*, *Myrmecridium*, *Pichia*, *Monascus*, *Acremonium*, *Schizangiella*, *Microidium*, *Pyrenochaetopsis*, *Halomyces*, *Phaeosphaeria*, *Enterographa*, *Leptospora*, and *Botrytis* (Fig. 5A and Table S3). Meanwhile, 30 differentially abundant fungal genera were identified in the rhizosphere soil, among which 25 increased and 5 decreased in abundance in the DT-8-treated wheat. The top 10 genera, *Scedosporium*, *Heydenia*, *Sporobolomyces*, *Mortierella*, *Shiraia*, *Leptospora*, *Psathyrella*, *Cyphellophora*,

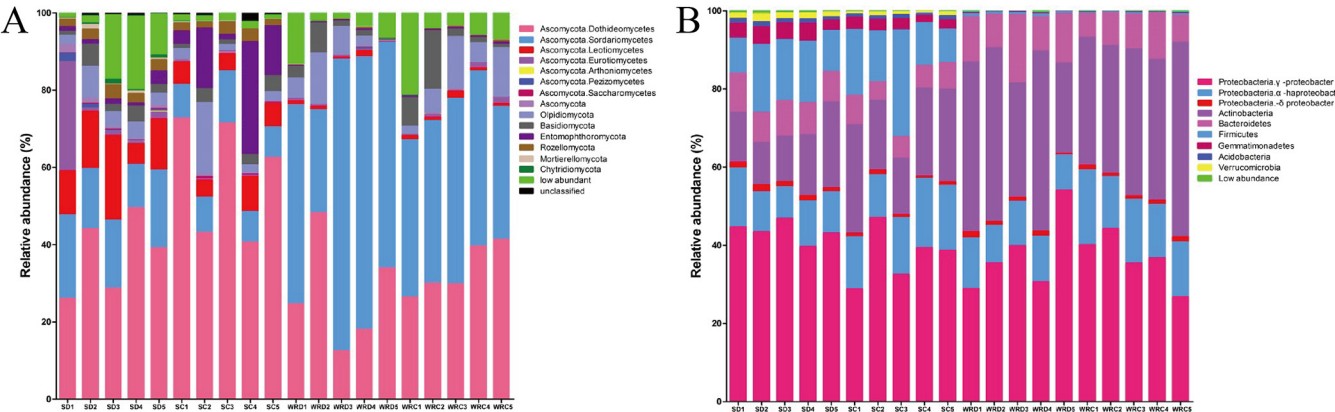

**FIG 4** Microbiome composition at the phylum level of rhizosphere soil and root samples. (A) Relative sequence abundance of fungal phyla associated with the rhizosphere soil and the roots. The phylum Ascomycota has been described as 7 OTUs at the class level (Dothideomycetes, Sordariomycetes, Leotiomycetes, Eurotiomycetes, Arthoniomycetes, Pezizomycetes, and Saccharomycetes). (B) Relative sequence abundance of bacterial phyla associated with the rhizosphere soil and the roots. The phylum *Proteobacteria* has been described as 3 OTUs at the subclass level (*Alpha-*, *Delta-*, and *Gammaproteobacteria*). Biological replicates are displayed in separate stacked bars. Major contributing phyla are displayed in different colors, and minor contributing phyla are grouped and displayed in gray. Low-abundance taxonomic groups with less than 1% of total reads across all samples are highlighted in green.

*Chaetomium*, and *Preussia*, showed higher abundance in DT-8-treated samples, while *Schizangiella*, *Pyrenochaetopsis*, *Myrmecridium*, *Genolevuria*, and *Zymoseptoria* were abundantly present in nontreated samples (Fig. 5B and Table S4).

For bacterial communities, 68 differentially abundant bacterial genera were identified in the root, including 35 genera with increased abundance, and 33 genera had decreased abundance in the roots of DT-8-treated wheat. The top 10 genera, *Pseudorhodoferax*, *Bryobacter*, *Frondihabitans*, *Enterobacter*, *Ralstonia*, *Hydrogenophaga*, *Fictibacillus*, *Delftia*, *Ellin6067*, and *Parafrigoribacterium*, showed higher abundance in the roots of DT-8-treated wheat but lower abundance in those of the control. Dominant members of the nontreated wheat root samples were *Larkinella*, *Belnapia*, *Aquipuribacter*, *Runella*, *Virgisporangium*, *Simplicispira*, *Chthoniobacter*, *Polynucleobacter*, *Actinotalea*, *Planococcus*, etc. (Fig. 5C and Table S5). Interestingly, the relative abundances of many well-known plant growth-promoting rhizobacteria (PGPR), including *Bacillus*, *Fictibacillus*, *Paenibacillus*, *Enterobacter*, and *Phyllobacterium*, were higher in the DT-8 treatment wheat roots.

In addition, 67 differentially abundant genera were identified in the rhizosphere soil, including 36 genera with increased abundance, and 31 genera had decreased abundance in the rhizosphere soil of DT-8-treated wheat. *Mucilaginibacter*, *Sanguibacter*, *Dechloromonas*, *Mesotoga*, *Nannocystis*, *Parviterribacter*, metagenome, *Lacibacter*, *Herminiimonas*, *Clostridium*, etc. showed higher abundance in the rhizosphere soil of the DT-8-treated wheat but lower abundance in that of the control. Dominant members of the nontreated wheat rhizosphere soils were *Aureimonas*, *Rubellimicrobium*, *Corallococcus*, *Asticcacaulis*, *Caulobacter*, *Gryllotalpicola*, *Hartmannibacter*, *Aquicella*, *Ornithinicoccus*, *Solibacillus*, etc. (Fig. 5D and Table S6). Notably, well-known biocontrol microbes, including *Brevibacillus*, *Microbacterium*, *Paenibacillus*, *Bacillus*, and *Lecanicillium*, are enriched in the rhizosphere soil of the DT-8-treated wheat.

**Venn figure comparison.** To provide a complete overview of the operational taxonomic unit (OTU) distribution within different groups, the number of OTUs was calculated. For fungal communities, a total of 975 OTUs were generated. Approximately 35.9% (350 OTUs) of the total OTUs of fungi were commonly present in all samples, and 15.6% were exclusively found in rhizosphere soil compared to roots (17.8%). A total of 50.8% (495 OTUs) of total fungal OTUs were shared among the rhizosphere soils and roots of DT-8-treated wheat plants. Meanwhile, 47.9% (227 OTUs) of total fungal OTUs were shared among the roots and corresponding rhizosphere soils of nontreated plants. In addition, 59 specific OTUs were separately identified in DT-8-treated wheat plant rhizosphere soil, compared to the nontreated control (28), DT-8-treated wheat roots (20), and nontreated wheat roots (45) (Fig. 6A).

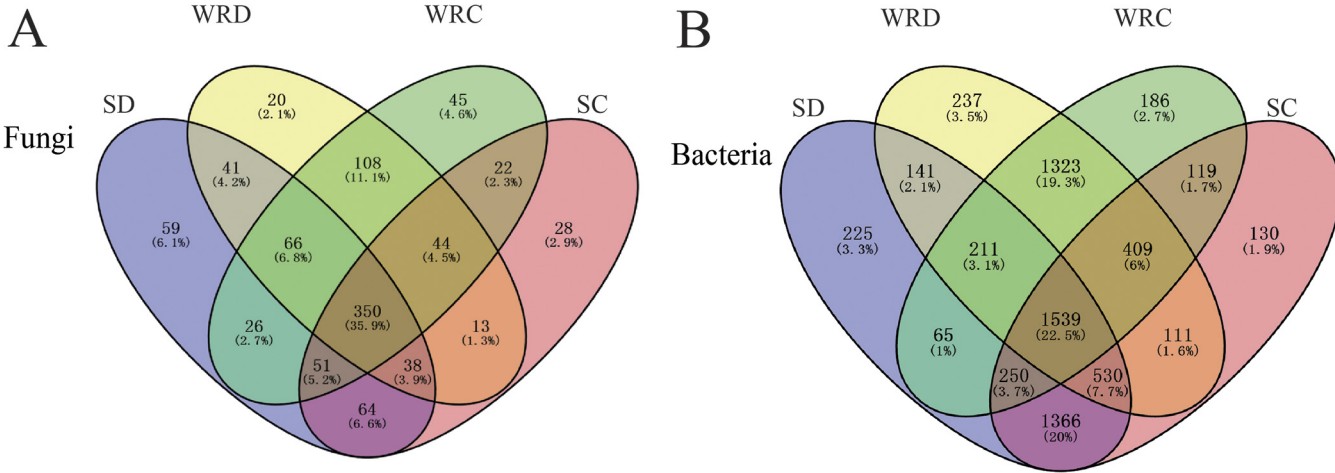

**FIG 5** Heatmap of the dominant fungi and bacteria at the genus level in the rhizosphere and root samples. (A and B) The differentially abundant genera of fungi in roots and the rhizosphere soil samples. (C and D) The differentially abundant genera of bacteria in roots and the rhizosphere soil samples (FDR *P* values < 0.05).

In the case of bacteria, 6,842 OTUs were generated in rhizosphere soil and roots in the DT-8-treated and control groups. About 22.5% of the total OTUs of bacteria were commonly present in all samples, and 25.2% were exclusively found in rhizosphere soil between DT-8-treated and control plants compared to roots (25.5%). For DT-8-treated roots, 35.4% (2,421 OTUs) of the total bacterial OTUs were shared with the corresponding rhizosphere soil of the DT-8-treated wheat. Meanwhile, 33.9% (2,317 OTUs) of the

**FIG 6** The overlaps of the wheat-associated microbiome at the OTU level in all samples. (A) Fungal OTUs in the rhizosphere soil and root samples. (B) Bacterial OTUs in the rhizosphere soil and root samples.

total bacterial OTUs were shared with the corresponding rhizosphere soil in nontreated root samples. Nearly 3.3% (225 OTUs) were particularly occupied by DT-8-treated wheat corresponding rhizosphere soil, compared to the nontreated control (1.9%, 130 OTUs), DT-8-treated wheat roots (3.5%, 237 OTUs), and nontreated wheat roots (2.7%, 186 OTUs) (Fig. 6B).

**Indicator species analyses.** To support the analysis of variance (ANOVA) results at the OTU level and further ascertain the OTUs responsible for the observed community differentiation within the roots and the rhizosphere soil between DT-8-treated and control wheat plants, we used species indicator analyses to discover significant associations between OTUs and samples. Indicator analyses were performed on the complete community matrices (not only core OTUs) to uncover the effect possibly missed by the core OTU analysis. As a result of ITS sequencing, 47 indicator species from the top 600 OTUs are listed in Table S7, comprising 23 species from the rhizosphere soil of DT-8-treated wheat samples, 8 species from the rhizosphere soil of nontreated wheat samples, and 2 and 14 species from the roots of DT-8-treated and nontreated wheat samples, respectively. However, when we used a community matrix excluding OTUs with an average relative abundance of $\geq$1%, we found 2 indicator OTUs in the rhizosphere soil of DT-8-treated wheat samples (*Aspergillus* and *Helotiales* [$P < 0.001$]) and 1 in roots of the nontreated wheat samples (*Pseudocercospora* [$P < 0.01$]). Indicator species analysis of the bacterial community yielded 58 indicator species from the top 2,000 OTUs. These results show that 21 and 15 were distributed in roots and the corresponding rhizosphere soils of DT-8-treated wheat samples, respectively. In addition, 5 and 17 were distributed in the roots and corresponding rhizosphere soils of nontreated wheat samples, respectively. The complete lists of indicator OTUs and their corresponding indicator values can be found in Table S8.

## DISCUSSION

Previously, Trianum-P and some beneficial bacteria have been found to change the composition and diversity of fungal communities of the strawberry phyllosphere and bacterial communities of the chamomile rhizosphere, respectively (14, 15). In this study, we found that systemically endogenous growth of *S. sclerotiorum* strain DT-8, a mycovirus-mediated hypovirulent strain, affected the composition and diversity of microbial communities in wheat roots and rhizosphere soil using 16S rRNA gene and ITS-sequencing techniques. DT-8 treatment increased the relative abundance of possible PGPR and biocontrol agents in the rhizosphere soil of wheat plants, whereas it decreased the relative abundance of fungi in wheat roots, which may be one of the major reasons for the growth promotion and increased yield of wheat treated with DT-8 to suppress FHB .

In this study, the number of observed fungal OTUs in the treated wheat roots was significantly lower than that of the nontreated control, but the observed bacterial OTUs were not significantly different. Endophytic fungi typically live inside plants without harming them, and their growth in plants is controlled by both sides, the fungus and the plant host (16). The reciprocal interplay between the microbiota and the immune system plays a critical role in maintaining microbial homeostasis (17). On one hand, endophytic fungi could enhance plant resistance to biotic stresses via induced systemic resistance (ISR) of the host plant (18, 19). In previous research, we found that strain DT-8 modified the expression of wheat genes involved in pattern-triggered immunity (PTI), abscisic acid (ABA), and jasmonic acid (JA) metabolism related to disease resistance (13). In addition, Zhang et al. reported that *S. sclerotiorum* strain DT-8 grows endophytically in rapeseed plants and significantly upregulates some key genes that respond to biotic and abiotic stresses (20). On another hand, the growth of endophytic fungi is usually limited by inducing the surrounding plant cells to deposit cell wall material and produce phenolic compounds and to avoid an additional burden to the host plant (16). Furthermore, various microorganisms compete for similar niches and nutrients in plant roots, and widespread competitive microbe-microbe interactions between members of fungal communities and bacterial communities are likely to

occur (17). Hence, we speculate that endophytic growth of *S. sclerotiorum* in wheat roots could enhance the ISR of wheat plants and enhance itself to compete with other microorganisms, especially fungal communities in wheat roots, thereby reducing the diversity of fungi there.

The homeostatic balance between both microbe-microbe and host-microbe is critical for a healthy host-microbiota relationship (21). In our study, DT-8 treatment significantly increased fungal and bacterial diversity in wheat rhizosphere soil. Plant roots exude an enormous range of potentially valuable small-molecular-weight compounds into the rhizosphere, including the secretion of ions, free oxygen and water, enzymes, mucilage, and a diverse array of carbon-containing primary and secondary metabolites (22). In general, plants drive the composition and structure of rhizosphere bacterial communities through root exudates (23, 24). In turn, rhizosphere microorganisms can promote the overall health of plant species by promoting crop growth and participating in root surface defense (25, 26). We currently do not know the underlying mechanisms responsible for *S. sclerotiorum* affecting the diversity of bacteria and fungi. However, it is plausible that *S. sclerotiorum* may have direct or indirect effects on the root exudates of the wheat plants. This hypothesis needs to be further confirmed with experiments.

Maintaining root activity is critical for enhancing N uptake efficiency (NUpE) during the entire growing season and can increase grain N content and N use efficiency (NUE) during the grain-filling period (27, 28). Microbes play an essential role in nitrogen turnover in the root zone, especially the root-associated bacteria related to the utilization of nitrogen elements that are critical for plant nitrogen uptake (9, 29). In this study, we found that some bacterial genera related to root activity were "activated" in the DT-8 treated wheat rhizosphere soil, such as *Clostridium* (30), *Brevibacillus* (31), *Nannocystis* (32), *Paenibacillus* (33), and *Microbacterium* (34). In addition, DT-8-treated wheat rhizosphere soil recruited a higher proportion of *Nitrospirae* bacteria related to the nitrogen cycle, indicating that the nitrogen transformation process is probably more efficient in the root environment of DT-8-treated wheat plants than in the control. Above all, these results may explain the previous results of higher nitrogen content in DT-8-treated wheat flag leaves than in the control.

Interestingly, *Allorhizobium* was detected in both the DT-8-treated and nontreated wheat root samples, and the relative abundance in the former was much lower than that in the latter. In the previous studies, *Rhizobia* were found to live in nonleguminous plants, for example, in rice roots, but they could not induce nitrogen fixation (35–37). Furthermore, *Mesorhizobium huakuii* was found to invade *Plasmodiophora brassicae*-infected rapeseed root, fix nitrogen in *P. brassicae*-infected plants, alleviate clubroot symptoms, and promote the growth of diseased rapeseeds (38). However, the role of rhizobia is still unclear in the wheat root and needs further study.

PGPRs are a group of microbes that colonize plant roots and enhance the adaptive capacity of host plants either directly or indirectly (39, 40). PGPR benefits plants through biofertilization, stimulation of root growth, rhizoremediation, and plant stress control (39). Previously, we found that *S. sclerotiorum* promotes wheat yields by 4 to 18% by modifying the expression of wheat genes involved in photosynthesis and indole-3-acetic acid (IAA) (13). In this study, the relative abundances of many well-known PGPR, including *Bacillus* (41), *Fictibacillus* (42), *Paenibacillus* (43), *Enterobacter* (44), and *Phyllobacterium* (45), were higher in the DT-8 treatment group wheat roots. This result may offer another explanation for the increased yield of wheat plants after being treated with DT-8.

Biological control refers to the intentional use of introduced or resident microorganisms to suppress one or more plant pathogens and presents an environmentally friendly approach to increasing world crop production (5, 46). Induced systemic resistance (ISR) emerged as an important mechanism by commensal bacteria and fungi in the rhizosphere of plants for enhanced defense against a broad range of pathogens (47). Previously, we found that *S. sclerotiorum* provided protection against FHB and wheat rust by modifying the expression of wheat genes related to disease resistance

and proposed that the resistance invoked by strain DT-8 is likely to be broad spectrum. In this study, we also found that DT-8-treated wheat root and rhizosphere soil recruited more potential biological control bacteria and fungi able to invoke ISR, such as *Brevibacillus* (48, 49), *Microbacterium* (50, 51), *Paenibacillus* (52–55), *Bacillus* (36, 56–58), and *Lecanicillium* (59, 60). Therefore, we suggest that strain DT-8 not only activates ISR via seed treatment but also stimulates the wheat root system or other microbes to secrete second metabolites and recruit more biocontrol resources.

The purpose of this study was to discover the wheat root and corresponding rhizosphere microbial communities treated by *S. sclerotiorum*. Previously, we demonstrated that the beneficial effect of *S. sclerotiorum* on wheat was not strain dependent and did not depend on full virulence on susceptible hosts. Furthermore, given the potential risks associated with a virulent *S. sclerotiorum* strain to dicotyledonous host plants such as oilseed rape and soybean in natural ecosystems, as well as the stability of SsHADV-1 in strain DT-8 and its potential as a "plant vaccine" against sclerotinia disease in oilseed rape, we only used the mycovirus-mediated hypovirulent DT-8 strain, which is safe for dicotyledons such as oilseed rape and monocotyledons such as wheat in this study. Our data suggest that *S. sclerotiorum* colonizes wheat roots to enrich the taxon-related nitrogen metabolism, potential PGPR, and biocontrol agents in wheat roots and the corresponding rhizosphere microbial communities, enhance plant disease resistance, and promote plant growth under natural field conditions. The results of the interaction between the biocontrol agent (*S. sclerotiorum*) and the microbiota of root and rhizosphere could provide a perspective for the screening and utilization of beneficial microorganisms and further reduce chemical pesticide utilization and increase crop productivity.

## MATERIALS AND METHODS

**Plant and fungal materials, maintenance, and preparation.** *S. sclerotiorum* strain DT-8, originally isolated from a *Sclerotium* isolate collected in diseased rapeseed, is a hypovirulent strain infected by a DNA virus (61). Strain DT-8 was grown on potato dextrose agar (PDA) or potato dextrose broth (PDB) at 20°C and stored on PDA slants at 4°C.

Seeds of the winter wheat cultivar Zheng 9023 were purchased from the commercial seed market in Wuhan City, China. Wheat seeds were washed with tap water and surface-sterilized with 0.5% NaClO for 10 min and then washed three times with sterilized water. Surface-sterilized seeds were soaked in sterile water for 4 h and then collected and blotted dry. Meanwhile, strain DT-8 was cultured in 250-mL flasks in a shaker at 20°C for 4 days, and the hyphal fragment suspensions were then used to inoculate the prepared seeds (100 mL of hyphal fragment suspension/kg wheat seeds), after which the inoculated seeds were ventilated to dry. Wheat seeds soaked in sterilized water only were used as a control.

**Wheat cultivation, sample collection, and plant traits.** A wheat field located in Ezhou, Hubei Province, China, was selected for the experiment in late October 2018. Each wheat treatment was grown in a plot of 30 m$^2$ (15 m by 2 m) with 30 cm between plots. Field management was conducted as per normal farming practice, except that no fungicides were applied. Five replicates were selected for each group at the initial bloom stage of the wheat, and each replicate consisted of three pooled individuals for sequencing. Roots were shaken to remove the loosely adhering soil and then washed with phosphate-buffered saline (PBS) buffer until no visible soil particles remained, and the soil in PBS buffer was used as rhizosphere samples. Further, roots were sterilized with 70% ethyl alcohol (EtOH) for 2 min and then in 0.5% sodium hypochlorite for 30 min, thoroughly washed with sterile distilled water 3 times, and used for the analysis of microbes residing inside roots.

**Detection of nitrogen.** Wheat flag leaves and soil samples were collected from DT8-treated and control experimental plots at the initial bloom stage. Six replicates were included for each group, and each replicate consisted of five pooled individuals. Dry weights were determined after oven-drying at 70°C to a constant weight. Nitrogen concentrations of wheat flag leaves were determined by standard micro-Kjeldahl digestion, distillation, and titration to calculate the total nitrogen content. Methods for the determination of nitrogen were performed as previously described (62).

**DNA extraction and amplification.** Wheat roots and the corresponding rhizosphere soil samples were analyzed for bacterial 16S rRNA gene and fungi ITS gene profiling by Illumina sequencing. The DNA for each sample was extracted with a FastDNA spin kit (MP Biomedicals) according to the protocol for the isolation of DNA. The concentration of extracted DNA was quantified using a Qubit 2.0 fluorometer (Invitrogen, Carlsbad, CA, USA) and subsequently diluted to 3.5 ng/$\mu$L and used in a two-step PCR amplification protocol. In the first step, V4 to V7 of bacterial 16S rRNA (799F/5′-AACMGGATTAGATACCCKG-3′; 1193R/5′-ACGGGCGGTGTGTRC-3′) and fungal ITS1 (ITS1/5′-CTTGGTCATTTAGAGGAAGTAA-3′; ITS2/5′-GCTGCGTTCTTCATCGATGC-3′) were amplified. Under a sterile hood, each sample was amplified in triplicate in a 25-$\mu$L reaction volume containing TransStart buffer (2.5 U/$\mu$L PFU), deoxynucleoside triphosphates (dNTPs; 2.5 mM each), 0.3 $\mu$M forward and reverse primers, TransStart *Taq* DNA polymerase (2.5 U/$\mu$L) and 20 ng DNA. The PCRs were conducted using the following program: 94°C for 2 min, 94°C for 30 s, 55°C for 30 s, 72°C for 30 s, and 72°C for 10 min for 25

cycles. Afterward, single-stranded DNA and proteins were digested by adding 1 $\mu$L of Antarctic phosphatase, 1 $\mu$L exonuclease I, and 2.44 $\mu$L Antarctic phosphatase buffer (New England BioLabs GmbH, Frankfurt, Germany) to 20 $\mu$L of the pooled PCR product. Samples were incubated at 37℃ for 30 min, and enzymes were deactivated at 85℃ for 15 min. Samples were centrifuged for 10 min at 4,000 rpm, and 3 $\mu$L of this reaction mixture was used for a second PCR, prepared in the same way as described above using the same protocol but with the number of cycles reduced to 10. PCR quality was controlled by loading 5 $\mu$L of each reaction mixture on a 1% agarose gel and confirming that no band was detected within the negative control. Afterward, the replicated reactions were combined and purified: (i) amplicons were loaded on a 1.5% agarose gel and run for 2 h at 80 V; bands with the correct size of ~500 bp were cut out and purified using the NanoDrop 2000C device (Thermo Scientific); (ii) fungal samples were purified using Agencourt AMPure XP beads. The DNA concentration was again fluorescently determined, and 30 ng DNA of each of the barcoded amplicons was pooled in one library per microbial group. Each library was then purified and reconcentrated twice with Agencourt AMPure XP beads, and 100 ng of each library was pooled together. Paired-end Illumina sequencing was performed in-house using the NovaSeq sequencer and custom sequencing primers. Illumina NovaSeq sequencing was performed by a commercial company (Personalbio Technology, Shanghai, China). Sequenced data for 16S rRNA and ITS2 have been submitted to NCBI SRA under BioProject no. PRJNA601289 and PRJNA545802.

**PCR detection.** To test the SsHADV-1 in all wheat root samples, the coat protein gene (*CP*) fragment of SsHADV-1 was amplified using specific primers (CP-F1: 5′-GGAGCATCCTCAACACGACATC-3′ and CP-R1: 5′-TACGAAGAAGGTCGGACGCC-3′). A total volume of 25 $\mu$L PCR mixture contained 2.5 $\mu$L 10$\times$ buffer (Mg$^{2+}$) (New England Biolabs), 0.5 $\mu$L 10 mM dNTP mix (New England Biolabs), 0.5 $\mu$L of each primer (10 $\mu$M), 0.2 $\mu$L *Taq* DNA polymerase (5 U/$\mu$L) (New England Biolabs), 19.8 $\mu$L nuclease-free water, and 1 $\mu$L genomic DNA (10 to 100 ng). The conditions for PCR amplification included a denaturation step at 95℃ for 5 min and 30 cycles of 94℃ for 30 s, 58℃ for 45 s, and 72℃ for 5 min.

**Statistical analyses.** The raw data were obtained from the Illumina sequencing platform, including low-quality reads and adaptor sequences. Sequences were preprocessed, quality filtered, and analyzed using Quantitative Insights into Microbial Ecology 2 (QIIME 2) version 2019.4 (63). The DADA2 software package was used to control the sequence quality and remove chimeras with the "consensus" method (64). Taxonomic assignment of 16S rRNA gene and ITS fragment representative sequences was performed based on the Greengenes database (65) and the UNITE database (66).

After discarding the no-target OTUs and low-abundance OTUs (<5 total counts) (67), the rarefaction curves of the 16S rRNA gene and ITS-sequencing data were created using QIIME 2. $\alpha$-Diversity analysis was performed with QIIME 2. For fungal communities and bacterial communities, the Chao1 index and Simpson index were used to evaluate the richness and evenness, respectively; Shannon's diversity index was used to calculate the diversity. The Kruskal-Wallis test was used to analyze the statistical differences in $\alpha$-diversity. For $\beta$-diversity analyses, nonmetric multidimensional scaling (NMDS) and hierarchical clustering (unweighted pair-group method with arithmetic means) were performed using the Bray-Curtis dissimilarity matrices and were visualized using the R package vegan (R version 3.1).

The Kruskal-Wallis test was used to detect the differential relative abundance of bacterial and fungal OTUs among samples using mothur without *P* value adjustment. The OTU statistical information of all samples was rendered using the Venny figure. The difference in the microorganism population between DT-8-treated and control wheat roots and the corresponding rhizosphere soil samples was determined by performing community cluster analysis using Euclidean distance, which was performed using the heatmap function of TBtools (version 1.6) (68).

The data are expressed as the mean $\pm$ standard deviation (SD) and were analyzed using SPSS version 19 statistical software. Significant differences between the two groups were evaluated by with two-tailed unpaired Student's *t* test for samples that were not normally distributed. Significant differences among three or more groups were evaluated by one-way ANOVA with Bonferroni's multiple-comparison tests. The level of significance was set at the following: *, $P < 0.05$; **, $P < 0.01$; ***, $P < 0.001$.

**Data availability.** All sequence reads generated in this study have been uploaded onto the NCBI GEO database with BioProject no. PRJNA601289 and PRJNA545802.

## SUPPLEMENTAL MATERIAL

Supplemental material is available online only.
**SUPPLEMENTAL FILE 1**, XLSX file, 0.01 MB.
**SUPPLEMENTAL FILE 2**, XLSX file, 0.01 MB.
**SUPPLEMENTAL FILE 3**, XLSX file, 0.01 MB.
**SUPPLEMENTAL FILE 4**, XLSX file, 0.01 MB.
**SUPPLEMENTAL FILE 5**, XLSX file, 0.02 MB.
**SUPPLEMENTAL FILE 6**, XLSX file, 0.02 MB.
**SUPPLEMENTAL FILE 7**, XLSX file, 0.01 MB.
**SUPPLEMENTAL FILE 8**, XLSX file, 0.01 MB.
**SUPPLEMENTAL FILE 9**, PDF file, 0.2 MB.

## ACKNOWLEDGMENTS

This work was financially supported by the National Key Research and Development Program of China (2022YFD1400100), the Fundamental Research Funds for the Central

Universities (SWU120061), the National Science Foundation of China (32130087), and the earmarked fund for CARS-12.

We also thank the reviewers for their valuable comments.

B.T. designed the research, performed all experiments, analyzed data, and cowrote the manuscript. Z.Q. and M.A.M. analyzed bioinformatics data. J.X. and J.C. analyzed data. D.J. and Y.F. designed and supervised the research and cowrote the manuscript.

We have no conflict of interest.

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
