## [Reviewer comments · Microbiology Spectrum]

Microbiology Spectrum

Schizotrophic *Sclerotinia sclerotiorum*-mediated root and rhizosphere microbiome alterations activate the growth and disease resistance in wheat

binnian tian, Zheng Qu, Mirza Abid Mehmood, Jiatao Xie, Jiaseng Cheng, Yanping Fu, and Daohong Jiang

Corresponding Author(s): Daohong Jiang, Huazhong Agricultural University

Review Timeline:

Submission Date:	March 8, 2023
Editorial Decision:	April 10, 2023
Revision Received:	April 21, 2023
Accepted:	May 3, 2023

Editor: Zhongxiong Lai

Reviewer(s): Disclosure of reviewer identity is with reference to reviewer comments included in decision letter(s). The following individuals involved in review of your submission have agreed to reveal their identity: Shi-Hong Zhang (Reviewer #2)

Transaction Report:

DOI: <https://doi.org/10.1128/spectrum.00981-23>

April 10, 2023

Dr. binnian tian
Southwest University
tiansheng road 2
chongqing
China

Re: Spectrum00981-23 (Schizotrophic Sclerotinia sclerotiorum-mediated root and rhizosphere microbiome alterations activate the growth and disease resistance in wheat)

Dear Dr. binnian tian:

Link Not Available

The ASM Journals program strives for constant improvement in our submission and publication process. Please tell us how we can improve your experience by taking this quick Author Survey.
my decision is minor revision.

Sincerely,

Zhongxiong Lai

Journals Department
Reviewer comments:

Reviewer #1 (Comments for the Author):

In this study, strain DT-8 could significantly increase the diversity of the fungal and bacterial community in rhizosphere soil, while the diversity of the fungal community was obviously decreased in the wheat root.

Specific comments:

1. L154-L156:"These results indicate that *S. sclerotiorum*-treatment significantly increased fungal and bacterial diversity index in wheat rhizosphere soil, whereas declining thefungal diversity" should be modified as "These results indicate that *S. sclerotiorum*-treatment significantly increased fungal and bacterial diversity index in wheat rhizosphere soil, whereas declining thefungal diversity in roots".

2. L200-L203: "a total of 19 differentially abundant fungal genera were observed in wheat roots, including 2 genera with increased abundance and 17 genera had decreased abundance in the root of DT-8-treated wheat." should be modified as "a total of 18 differentially abundant fungal genera were observed in wheat roots, including 2 genera with increased abundance and 16 genera had decreased abundance in the root of DT-8-treated wheat."
3. L203-L204: "Rhizoctonia and Sclerotinia showed higher abundance in the rhizosphere soil of the DT-8-treated wheat, while lower abundance in the control." should be modified as "Rhizoctonia and Sclerotinia showed higher abundance in the root of the DT-8-treated wheat, while lower abundance in the control."
4. L208: "Phaeosphaeria, Enterographa, Podospora, Leptospora and Botrytis (Fig. 5A and" there is no Podospora in Fig. 5A.
5. L238-L240: "Notably, well-known bio-control microbes, including Brevibacillus, Microbacterium, Paenibacillus, Bacillus, and Lecanicillium are enriched in the root and rhizosphere soil corresponding DT-8-treated wheat." should be modified as "Notably, well-known bio-control microbes, including Brevibacillus, Microbacterium, Paenibacillus, Bacillus, and Lecanicillium are enriched in the rhizosphere soil corresponding DT-8-treated wheat."
6. L241-L263: In this venn figure comparison, the OTUs numbers is inconsistent with the Fig. 6. It should be checked carefully.
7. L283-L284: Where is Supplementary Table 8?
8. L270-L271: "As a result of ITS sequencing, 57 indicator species from the top 600 OTUs are given in (Supplementary Table 7)," is it 57 or 47? Please check carefully.
9. L114-L117: Fig 1D and Fig 1E should be cited in this site.

Reviewer #2 (Comments for the Author):

The authors provided a logical and clear report on the rationale of the experiments and their results. The data are also well presented and the results support the authors conclusion. I don't have particular concerns about most of the experiments. To strengthen the manuscript, I suggest the authors present some previous results about the rhizosphere microbiome between mycovirus-mediated hypovirulent strain and non-virus infected strains in the introduction section. If the relevant results obtained previously, non-virus infected strains tested as control should also be discussed in discussion section.

Staff Comments:

Preparing Revision Guidelines

Please return the manuscript within 60 days; if you cannot complete the modification within this time period, please contact me. If you do not wish to modify the manuscript and prefer to submit it to another journal, please notify me of your decision immediately so that the manuscript may be formally withdrawn from consideration by Microbiology Spectrum.

Comments:

In this study, strain DT-8 could significantly increase the diversity of the fungal and bacterial community in rhizosphere soil, while the diversity of the fungal community was obviously decreased in the wheat root. However, there are some problems in the data in the manuscript. It need more modifications. It could be accepted after revised.

Specific comments:

1. L154-L156: “These results indicate that *S. sclerotiorum*-treatment significantly increased fungal and bacterial diversity index in wheat rhizosphere soil, whereas declining the fungal diversity” should be modified as “These results indicate that *S. sclerotiorum*-treatment significantly increased fungal and bacterial diversity index in wheat rhizosphere soil, whereas declining the fungal diversity in roots”.
2. L200-L203: “a total of 19 differentially abundant fungal genera were observed in wheat roots, including 2 genera with increased abundance and 17 genera had decreased abundance in the root of DT-8-treated wheat.” should be modified as “a total of 18 differentially abundant fungal genera were observed in wheat roots, including 2 genera with increased abundance and 16 genera had decreased abundance in the root of DT-8-treated wheat.”
3. L203-L204: “*Rhizoctonia* and *Sclerotinia* showed higher abundance in the rhizosphere soil of the DT-8-treated wheat, while lower abundance in the control.” should be modified as “*Rhizoctonia* and *Sclerotinia* showed higher abundance in the root of the DT-8-treated wheat, while lower abundance in the control.”
4. L208: “*Phaeosphaeria*, *Enterographa*, *Podospora*, *Leptospora* and *Botrytis* (Fig. 5A and” there is no *Podospora* in Fig. 5A.
5. L238-L240: “Notably, well-known bio-control microbes, including *Brevibacillus*, *Microbacterium*, *Paenibacillus*, *Bacillus*, and *Lecanicillium* are enriched in the root and rhizosphere soil corresponding DT-8-treated wheat.” should be modified as “Notably, well-known bio-control microbes, including *Brevibacillus*, *Microbacterium*, *Paenibacillus*, *Bacillus*, and *Lecanicillium* are enriched in the rhizosphere soil corresponding DT-8-treated wheat.”
6. L241-L263: In this venn figure comparison, the OTUs numbers is inconsistent with the Fig. 6. It should be checked carefully.
7. L283-L284: Where is Supplementary Table 8?
8. L270-L271: “As a result of ITS sequencing, 57 indicator species from the top 600 OTUs are given in (Supplementary Table 7),” is it 57 or 47? Please check carefully.
9. L114-L117: Fig 1D and Fig 1E should be cited in this site.

Point to point response

Comments from editor

Response #1: Thank you for processing the manuscript "Spectrum00981-23" and for the reviewer's comments. We have revised the manuscript carefully according to the comments, which we hope will meet with their approval.

Comments from reviewers:

Reviewer #1 (Comments for the Author):

In this study, strain DT-8 could significantly increase the diversity of the fungal and bacterial community in rhizosphere soil, while the diversity of the fungal community was obviously decreased in the wheat root.

Response #2: We are very grateful for these kind comments, and we have now modified the manuscript as appropriate to correct any deficiencies indicated by this referee as follows:

Specific comments:

1. L154-L156: "These results indicate that *S. sclerotiorum*-treatment significantly increased fungal and bacterial diversity index in wheat rhizosphere soil, whereas declining the fungal diversity" should be modified as "These results indicate that *S. sclerotiorum*-treatment significantly increased fungal and bacterial diversity index in wheat rhizosphere soil, whereas declining the fungal diversity in roots".

Response #3: Changed, please see line 156-158.

2. L200-L203: "a total of 19 differentially abundant fungal genera were observed in wheat roots, including 2 genera with increased abundance and 17 genera had decreased abundance in the root of DT-8-treated wheat." should be modified as "a

total of 18 differentially abundant fungal genera were observed in wheat roots, including 2 genera with increased abundance and 16 genera had decreased abundance in the root of DT-8-treated wheat."

Response #4: Changed, please see line 203-204.

3. L203-L204: "Rhizoctonia and Sclerotinia showed higher abundance in the rhizosphere soil of the DT-8-treated wheat, while lower abundance in the control." should be modified as "Rhizoctonia and Sclerotinia showed higher abundance in the root of the DT-8-treated wheat, while lower abundance in the control."

Response #5: Changed, please see line 205-207.

4. L208: "Phaeosphaeria, Enterographa, Podospora, Leptospora and Botrytis (Fig. 5A and" there is no Podospora in Fig. 5A.

Response #6: We apologize for this oversight. We have removed "Podospora" in the text in the revised version. Please see line 210.

5. L238-L240: "Notably, well-known bio-control microbes, including Brevibacillus, Microbacterium, Paenibacillus, Bacillus, and Lecanicillium are enriched in the root and rhizosphere soil corresponding DT-8-treated wheat." should be modified as "Notably, well-known bio-control microbes, including Brevibacillus, Microbacterium, Paenibacillus, Bacillus, and Lecanicillium are enriched in the rhizosphere soil corresponding DT-8-treated wheat."

Response #7: Changed, please see line 240-242.

6. L241-L263: In this venn figure comparison, the OTUs numbers are inconsistent with the Fig. 6. It should be checked carefully.

Response #8: We have carefully checked the description of the Venn figure, and corrected it in the revised version. Please see line 244-266.

7. L283-L284: Where is Supplementary Table 8?

Response #9: Supplementary Table 8 is now added in the revised version.

In the original manuscript it was probably the limitation of the number of Supplementary Tables that caused Supplementary Table 8 to be lost, we added Supplementary Table 8 in the revised version. please see Supplementary Table 8.

8. L270-L271: "As a result of ITS sequencing, 57 indicator species from the top 600

OTUs are given in (Supplementary Table 7)," is it 57 or 47? Please check carefully.

Response #10: Changed, it is 47. Please see line 274.

9. L114-L117: Fig 1D and Fig 1E should be cited in this site.

Response #11: Added, please see line 118-119. Since the contents of Figure 1D-E and Figure S1A-B were the same in the original manuscript, we deleted Figure S1 in the revised version. Please see line 784-786; 825-827.

Reviewer #2 (Comments for the Author):

The authors provided a logical and clear report on the rationale of the experiments and their results. The data are also well presented and the results support the authors conclusion. I don't have particular concerns about most of the experiments. To strengthen the manuscript, I suggest the authors present some previous results about the rhizosphere microbiome between mycovirus-mediated hypovirulent strain and non-virus infected strains in the introduction section. If the relevant results obtained previously, non-virus infected strains tested as control should also be discussed in discussion section.

Response #12:

We are very grateful for this kind comment. We have now added more information about our previous results about the beneficial effect between mycovirus-mediated hypovirulent strain and non-virus infected strain of *S. sclerotiorum* on wheat plants in the introduction section, please see line 97-102.

We also added this information in the discussion section in the revised version. Please see line 385-393.

Previously, we demonstrated that the beneficial effect of *S. sclerotiorum* on wheat was not strain dependent and did not depend on full virulence on susceptible hosts. Furthermore, given the potential associated risks of *S. sclerotiorum* virulent strain to dicotyledonous host plants such as rapeseed and soybean in natural ecosystems, as well as the stability of the SsHADV-1 in strain DT-8 and its potential as a "plant vaccine" against sclerotinia disease in rapeseed. We only used the mycovirus-mediated hypovirulent DT-8 strain that is safe for dicotyledons such as

rapeseed in this study. Therefore, we speculate that there is no significant difference in the effects of the strong virulent strain and the mycovirus-mediated hypovirulent strain on the wheat rhizosphere microbiome.

May 3, 2023

Dr. binnian tian
Southwest University
tiansheng road 2
chongqing
China

Re: Spectrum00981-23R1 (Schizotrophic Sclerotinia sclerotiorum-mediated root and rhizosphere microbiome alterations activate the growth and disease resistance in wheat)

Dear Dr. binnian tian:

Your manuscript has been accepted, and I am forwarding it to the ASM Journals Department for publication. You will be notified when your proofs are ready to be viewed.

Sincerely,

Zhongxiong Lai
Editor, Microbiology Spectrum
